# The Association between Circulating Branched Chain Amino Acids and the Temporal Risk of Developing Type 2 Diabetes Mellitus: A Systematic Review & Meta-Analysis

**DOI:** 10.3390/nu14204411

**Published:** 2022-10-20

**Authors:** Imran Ramzan, Arash Ardavani, Froukje Vanweert, Aisling Mellett, Philip J. Atherton, Iskandar Idris

**Affiliations:** 1Clinical, Metabolic and Molecular Physiology Research Group, MRC-Versus Arthritis Centre for Musculoskeletal Ageing Research, Royal Derby Hospital Centre, University of Nottingham, Derby DE22 6DT, UK; 2National Institute for Health Research (NIHR), Nottingham Biomedical Research Centre, Nottingham NG7 2UH, UK; 3Department of Nutrition and Movement Sciences, NUTRIM, School of Nutrition and Translational Research in Metabolism, Maastricht University, 6229 ER Maastricht, The Netherlands; 4School of Agriculture and Food Science, Agriculture and Food Science Centre, University College Dublin, Belfield, D04 V1W8 Dublin, Ireland

**Keywords:** BCAA, branch chain amino acid, leucine, isoleucine, valine, type 2 diabetes mellitus, T2DM, insulin resistance, obese, obesity, overweight, BMI

## Abstract

Introduction: Recent studies have concluded that elevated circulating branched chain amino acids (BCAA) are associated with the pathogenesis of type 2 diabetes mellitus (T2DM) and obesity. However, the development of this association over time and the quantification of the strength of this association for individual BCAAs prior to T2DM diagnosis remains unexplored. Methods: A systematic search was conducted using the Healthcare Databases Advance Search (HDAS) via the National Institute for Health and Care Excellence (NICE) website. The data sources included EMBASE, MEDLINE and PubMed for all papers from inception until November 2021. Nine studies were identified in this systematic review and meta-analysis. Stratification was based on follow-up times (0–6, 6–12 and 12 or more years) and controlling of body mass index (BMI) through the specific assessment of overweight cohorts was also undertaken. Results: The meta-analysis revealed a statistically significant positive association between BCAA concentrations and the development of T2DM, with valine OR = 2.08 (95% CI = 2.04–2.12, *p* < 0.00001), leucine OR = 2.25 (95% CI = 1.76–2.87, *p* < 0.00001) and isoleucine OR = 2.12, 95% CI = 2.00–2.25, *p* < 0.00001. In addition, we demonstrated a positive consistent temporal association between circulating BCAA levels and the risk of developing T2DM with differentials in the respective follow-up times of 0–6 years, 6–12 years and ≥12 years follow-up for valine (OR = 2.08, 1.86 and 2.14, *p* < 0.05 each), leucine (OR = 2.10, 2.25 and 2.16, *p* < 0.05 each) and isoleucine (OR = 2.12, 1.90 and 2.16, *p* < 0.05 each) demonstrated. Conclusion: Plasma BCAA concentrations are associated with T2DM incidence across all temporal subgroups. We suggest the potential utility of BCAAs as an early biomarker for T2DM irrespective of follow-up time.

## 1. Introduction

Type 2 diabetes mellitus (T2DM) and obesity are major causes of morbidity and mortality worldwide [1]. The World Health Organisation (WHO) estimates that, by the year 2035, 592 million patients with diabetes will exist worldwide [2]. The discovery of novel metabolites and biomarkers that reflect early changes in T2DM and obesity are crucial for both early diagnosis and the prevention of long-term complications. Indeed, a recent study determined that early detection and treatment of T2DM reduces the risk of cardiovascular morbidity and mortality [3].

Metabolomics has increasingly been implemented as an analytical approach to identify novel metabolites within a biological specimen [4,5] potentially facilitating their utilisation as biomarkers and novel drug therapy targets [6]. The utilisation of metabolomics in large epidemiological studies has also increased exponentially over the last decade. Specifically, the metabolic alterations associated with or preceding the development of T2DM makes this an attractive area of research [7]. Almost 50 metabolites have demonstrated an association with T2DM, but the most extensively researched group of metabolites are the branched chain amino acids (BCAAs) [8,9].

BCAAs consist of the amino acids isoleucine, leucine and valine. BCAAs are widely present in human food sources containing protein, such as meat and dairy food items, in addition to being supplemented by recreational and professional athletes in an isolated, supplement form [10,11,12,13]. Recent prospective studies have highlighted an association between elevated circulating levels of BCAAs and the development of T2DM [14,15,16]. In addition, studies that have successfully reduced circulating BCAA levels in humans have also demonstrated improvements in insulin sensitivity [17,18]. Consequently, elevated circulating levels of BCAAs could potentially serve as biomarkers for the development of T2DM [19,20,21]. However, systematic reviews that have investigated this association using cohort studies have provided conflicting results [16,22]. Furthermore, systematic reviews which analysed an association between individual (rather than total) BCAAs and risks of T2DM did not reach statistical significance [22]. Analysis derived from case–control studies would produce more robust findings due to the inherent reduction in the potential for confounding [23]. Further interpretation of these associations will also be strengthened by analysing total and individual BCAA levels at baseline and follow-up. [24,25]. A further ubiquitous issue within the literature is the presence of confounding variables, including body mass index (BMI) and ethnicity [26,27,28], with various studies implementing differing statistical procedures or variable lists for covariate adjustment [29,30,31]. Thus, any summative analysis of the relationship between BCAAs and T2DM should address these variables through a sufficiently structured study design.

Multiple in vitro and epidemiological studies have revealed variations in the effect of leucine, isoleucine and valine with various components of glucose metabolism [32,33,34]. Additionally, the relationship between BCAAs and their contribution to the development of T2DM phenotype with time has not been investigated previously [35]. The determination of temporality as an additional environmental contributor to the resulting T2DM phenotype, within the context of BCAA metabolism, is therefore of novel epidemiological and clinical interest. The identification of specific BCAA-temporal patterns may provide healthcare professionals with an evidence-based rationale to undertake timeframe and BCAA-specific assessments for the development of T2DM in at-risk patient groups, in order to facilitate early lifestyle and dietary interventions.

In this systematic review, an appraisal of the relationship between BCAAs and T2DM development is undertaken through case–control study data of patients with an overweight BMI status. In addition, we also aim to appraise this relationship with the effect of temporality, based on the duration of follow-up in the incorporated studies. To the best of our knowledge, no review has attempted to determine if the association between BCAAs and T2DM exhibits any variation based on follow-up time to diagnosis of T2DM. Finally, we also aimed to assess any differences in the effect size of each individual BCAA for each of the studied timeframes.

## 2. Materials and Methods

### 2.1. Protocol and Registration

The study protocol is registered in the International prospective register of systematic reviews (PROSPERO 2022; registration number CRD42022297132), with the entry registration occurring after the finalization of our research question and the implementation of our search strategy. The Preferred Reporting Items for Systematic Reviews and Meta-Analyses (PRISMA) reporting guidelines are provided (Appendix A) [36].

### 2.2. Eligibility Criteria

We exclusively sourced prospective case–control studies possessing pre- and post-follow up period individual plasma BCAA levels of at least one year in duration. The inclusion criteria, with respect to samples from within the case groups of the study cohorts, is patients that eventually developed non-insulin-dependent type 2 diabetes mellitus (T2DM) and a body mass index (BMI) of between 25 and 30 kg/m^2^. The control groups were matched for BMI. A full enumeration of our criteria for study inclusion and exclusion, with additional justifications, are provided through the population, intervention, comparison, and studies (PICOS) format (Appendix A).

### 2.3. Information Sources and Searches

MEDLINE and EMBASE databases were accessed through the NICE Healthcare Databases Advanced Search (HDAS). An additional search inquiry was performed through PubMed. Grey area literature sources came through Google Scholar and Scopus. These five databases constituted our primary sources. Studies that did not meet our inclusion criteria based on design, but nonetheless addressed the research question, were included into our secondary sources list, and were reviewed for additional studies through their references. The primary sources were all searched on 3 November 2021 and the secondary sources were reviewed on 13 January 2022. The full electronic search strategy preformed for MEDLINE, EMBASE, Pubmed, Scopus and Google Scholar are presented in (Appendix A).

### 2.4. Study Selection

Assessment of studies from primary sources performed by two independent researchers (I.R., F.V.) with a third (A.A.) serving as consensus former in the event of disagreement. The titles, abstracts and full texts of the included studies were evaluated after removal of any duplicates. Inter-observer agreement was found to be substantial (88.41%, Cohen’s κ 67%). Secondary sources were assessed by one independent researcher (A.A.). Study screening (assessment of manuscript suitability for inclusion based on our eligibility criteria) and eligibility (assessment for the presence of statistical data that met our eligibility criteria) were performed in parallel by two researchers (I.R., A.A.), and no disagreement was observed at either stage.

### 2.5. Data Collection Process

A single independent researcher (A.A.) created a data extraction form (Microsoft Excel spreadsheet). Thereafter, another independent researcher (A.M.) extracted the following information: first author, publication year, sample size, patient demographics, diabetic and BMI status, and main outcomes. In addition, any correlation statistics present in the studies that are present at the eligibility stage of screening were included for qualitative assessment and extracted into a separate excel sheet.

### 2.6. Intra-Study Risk of Bias Assessment

Intra-study bias: The Newcastle Ottawa tool (NOS) was used to assess the overall quality and reliability of the case–control studies that reached synthesis [37]. Assessment was performed by two independent researchers (I.R., F.V.) with a third (A.A.) providing consensus. The overall quality and extent of bias was ranked based on a summative count and assessed within the context of three domains: comparability of cases and controls, selection of cases and controls and exposure. These three domains were separated into 8 individual questions each receiving a single star except comparability which received two stars. Thus, each study was given a score of bias ranging from 7–9 stars “High quality”, 4–7 stars as “Fair quality” and 1–3 stars “Low quality”.

Inter-study bias: Should sufficient studies (n = 10) be encountered, publication bias will be assessed using Egger’s regression test through the Copenhagen Trial Unit’s Trial Sequential Analysis (TSA) software, with the generation of a funnel plot for visual inspection of distribution skew.

#### Summary Measures

The summary measure per data-point is the log–odds ratio (+-standard error), with the measure of effect reported as the inverse variance (IV) with 95% confidence intervals (CI) and a measure of significance (*p* < 0.05).

### 2.7. Synthesis of Results

The inverse-variance method and random effects model was selected due to anticipated imprecision. The effect size (reported as a Z-score) and study heterogeneity (reported as an I^2^ and Tau statistic) are reported. Heterogeneity is interpreted as advocated by Cochrane [38]. Should the original manuscripts present numeric correlation data, these were incorporated without further transformation or synthesis and are reported with their associated measurements of statistical significance (*p* < 0.05). Temporal groupings were defined on a continuous scale (follow-up duration in years) based on the requirement for at least three datapoints to be represented in each grouping, with the duration represented by each temporal grouping defined post hoc following allocation.

### 2.8. Risk of Bias across Studies

Inter-study bias: Should a sufficient number of studies (n = 10) be encountered, publication bias will be assessed using Egger’s regression test through the Copenhagen Trial Unit’s Trial Sequential Analysis (TSA) software, with the generation of a funnel plot [39].

### 2.9. Additional Analyses

In order to appraise the decision to include datapoints from studies with estimations of effect size through differing covariates (Table 1), sensitivity analysis was performed through all available datapoints for each of the amino acids and irrespective of follow-up duration. Herein, only those studies presenting effect size estimations through the inclusion of less than five covariates (Table 1) were retained for pooled estimation of effect size for each assessed amino acid and follow-up duration. In accordance with Cochrane guidelines [40], the resulting estimates are presented in tabulated form.

## 3. Results

### 3.1. Study Selection

During the identification phase, three hundred and five studies were discovered; after systematically appraising the studies, nine reached synthesis (Appendix A) [30,31,32,42,43,44,45,46,47]. A total of 29 studies at the eligibility stage were excluded based on inappropriate study design or an absence of usable data (Appendix A).

### 3.2. Study Characteristics

The study characteristics can be found in Table 1, they comprised of nine independent studies reporting data from 4313 T2DM patients and 10078 healthy controls [29,30,31,41,42,43,44,45,46]. All the included studies were case–controls studies, the control group comprised of non-diabetic volunteers. In the case group, all volunteers had laboratory-confirmed (either fasting glucose or glycated haemoglobin) T2DM and were not currently undertaking any T2DM medication. The average profile of the case cohort consists of participants with an age of 56.25 years, a BMI of 28.37 and a bias towards male biological sex (with 63.15% of participants across the synthesized studies being represented by men). The included studies have a publication date of between 2011 and 2018 representing the most recent and up-to date BCAA research [29,30,31,41,42,43,44,45,46]. The outcome of interest was reported as an odds ratios (OR) in eight of the included studies with a single study reporting as a β coefficient. The patient demographics across the cases and the controls ranged from 48.5 ± 13 to 69.5 ± 2.1 for age and were predominantly female, with mixed ethnicity or nationality status. Follow up time ranged from 4.7 to 20 years (Table 1).

### 3.3. Risk of Bias within Studies

All of the nine included studies received a ‘high quality’ score after assessment of the studies using the NOS scale. As per the NOS guidelines studies rewarded with 7–9 stars are considered as low risk of bias, high quality studies. All the included studies received this status with two of the studies receiving the maximum nine stars [41,44]. (Appendix A).

### 3.4. Results of Individual Studies

For this systematic review and meta-analysis, we only focused on the BCAAs and their metabolites. Since their discovery in the development of T2DM, the BCAAs are the most extensively and most consistently researched set of amino acids. The results of our meta-analysis performed for the three individual BCAAs are provided below (Table 2).

### 3.5. Valine

Within the zero-to-six year subgroup, three cohorts were incorporated for assessment (Palmer et al., 2015, Stancáková et al., 2012, Lu et al., 2016), with all demonstrating a positive association (OR = 2.08–2.56) (Figure 1) [29,30,45]. The overall OR was calculated to be 2.08 (95% CI = 2.04–2.12, *p* < 0.00001) (Figure 1). Although an assessment of overall heterogeneity was equivocal (I^2^ = 0%, *p* = 0.89) (Figure 1), a substantially wide 95% CI was observed with one cohort (Palmer et al., 2015, 0.49–13.28) (Figure 1) [29].

Four cohorts (Floegel et al., 2013, Ottosson et al., 2018, Shi et al., 2018, Wang-Sattler et al., 2012) were sourced for data in the subgroup with a follow-up of more-than-six, but less-than-twelve year follow-up period (Figure 1) [31,41,43,44]. A positive association was observed in all (OR = 1.27–2.25), with an overall OR of 1.86 (95% CI = 1.28–2.68, *p* = 0.001) (Figure 1). Subgroup heterogeneity was found to be considerable (I^2^ = 0%, *p* = 0.89) (Figure 1).

Four cohorts sourced from two separate studies (Tillin et al., 2015; European and South Asian, Wang et al., 2011; Framingham and Malmo datasets) represented the subgroup consisting of at least twelve years of follow-up, with each demonstrating demonstrated positive associations (OR = 2.08–2.41), resulting in an overall OR of 2.14 (95% CI = 1.81–2.53, *p* < 0.00001) (Figure 1) [42,46]. An overall heterogeneity assessment was also equivocal (I^2^ = 0%, *p* = 0.98), with substantially wide 95% CI observed in the Malmo cohort data from Wang et al., 2011 (Figure 1).

### 3.6. Leucine

Three cohorts (Palmer et al., 2015, Stancáková et al., 2012, Lu et al., 2016) represented the zero-to-six year subgroup, where all presented a positive association (2.10–2.51) and a resulting overall OR (2.10, 95% CI = 2.02–2.18, *p* < 0.00001) (Figure 2) [30,31,46]. A further three cohorts (Ottosson et al., 2018, Shi et al., 2018, Wang-Sattler et al., 2012) were present in the more-than-six, but less-than-twelve year follow-up subgroup (Figure 2) [31,41,44]. Similarly, each cohort presented a positive association (OR = 2.20–2.32) and a combined OR of 2.25 (95% CI = 1.76–2.87, *p* < 0.00001) (Figure 2). 

Within the subgroup consisting of at least twelve years’ follow-up, the same four cohorts were incorporated (Figure 1), with all exhibiting a positive association (OR = 2.12–2.32) (Figure 2). This result was reflected in the overall OR (2.25; 95% CI = 1.76–2.87, *p* < 0.00001) (Figure 2). Heterogeneity assessments for each of the above-described three temporal subgroups was equivocal (I^2^ = 0, *p* > 0.05 in each) (Figure 2).

### 3.7. Isoleucine

Once more, Palmer et al., 2015, Stancáková et al., 2012 and Lu et al., 2016 provided datapoints for the zero-to-six year follow-up period subgroup, with all demonstrating a positive association (OR = 2.12–2.51), as reflected in the overall subgroup value (OR = 2.12, 95% CI = 2.00–2.25, *p* < 0.00001) (Figure 3) [29,30,45]. As observed with the same subgroup present within the analysis for valine (Figure 3), an assessment of overall heterogeneity was equivocal (I^2^ = 0%, *p* = 0.94) (Figure 3), with Palmer et al., 2015 producing a substantially wide 95% CI (0.49–12.47) (Figure 3) [29]. Four cohorts from the same studies that were present in the analysis of the valine output were present for isoleucine (Figure 1 and Figure 3). Furthermore, all revealed a positive association (OR = 1.30–2.34), as reflected in the resulting overall subgroup OR (1.90, 95% CI = 1.27–2.84, *p* = 0.002) (Figure 3) [31,41,43,44]. An assessment for heterogeneity demonstrated a considerable estimate (I^2^ = 82%, *p* = 0.002) (Figure 3).

As present in valine and leucine, four data-points from two studies (Tillin et al., 2015 and Wang et al., 2011) were present in the subgroup with at least twelve years follow-up, where all demonstrated a positive association (OR = 2.12–2.34) and overall subgroup OR (2.16, 95% CI = 1.82–2.56, *p* < 0.00001) (Figure 3) [42,46]. Heterogeneity assessment was, once more, equivocal (I^2^ = 0%, *p* = 0.99) (Figure 3).

### 3.8. Synthesis of Results

#### Risk of Bias across Studies

Unfortunately, as only nine studies reached the synthesis stage, an assessment of publication bias was not possible in this instance.

### 3.9. Sensitivity

Palmer et al., 2015, Lu et al., 2016, Ottosson et al., 2018 and Wang et al., 2011 were found to have published association estimates with models adjusting for less than five covariates for each of the BCAAs, resulting in their incorporation for sensitivity analysis (Table 1). The resultant output validates the rationale to include all published association estimates, irrespective of their model adjustment status (Table 3).

## 4. Discussion

### 4.1. Summary of Evidence

Numerous prospective studies have consistently reported an association between circulating BCAA concentrations and development of T2DM [21,28,47]. Our meta-analysis demonstrated all three BCAAs exhibit a positive association with the development of T2DM (OR = 2.01–2.10, *p* < 0.00001). Our findings are in accordance with those of Guasch-Ferre 2016 et al., who also demonstrated a positive association between BCAAs and incident T2DM, with a pooled risk ratio (RR) for isoleucine, leucine and valine of 1.36 (95% CI 1.24–1.48), 1.36 (1.17–1.58), and 1.35 (1.19–1.53), respectively [16].

In addition, this is the first meta-analysis to evaluate the temporal association between BCAAs and risk of developing T2DM. The findings of our meta-analysis confirm that, across each individual BCAA, a significant and consistent risk for development of T2DM was observed at each time frame, prior to incident T2DM (Figure 1, Figure 2 and Figure 3). These positive associations between BCAAs and T2DM at variable period prior to incident T2DM suggests that elevated BCAA concentrations could potentially predict the onset of T2DM years before the clinical T2DM symptoms or sequelae of T2DM may manifest. The findings of our systematic review and their potential utility as biomarkers for T2DM is in agreement with multiple metabolomic studies that have confirmed BCAAs exhibit sufficient statistical association with T2DM to serve as risk biomarkers [16,20,42].

In addition, we observed a variation in association between the BCAAs and T2DM with valine providing the lowest overall OR (2.01, *p* < 0.00001) and leucine the highest (2.10, *p* < 0.00001) (Figure 1, Figure 2 and Figure 3). Similarly, this association was also observed in a meta-analysis by Y Sun, 2019 et al., who demonstrated an increased association between leucine and T2DM, with valine producing the weakest association of the BCAAs, where their RRs were 1.40 (95% CI 1.27–1.44) and 1.26 (95% CI 1.18–1.34), respectively [22]. Individually, the strength of each BCAA for T2DM varies between studies. In vitro studies have demonstrated leucine to exhibit a greater affinity in activating mTORc1 compared to either isoleucine or valine which is a pathway suggested to be involved in the development of insulin resistance [48,49] Indeed, the studies included in this review reported differing strengths in association for T2DM for each individual BCAA. Y Lu, 2016 et al., and ND Palmer 2015 et al., reported a higher OR for Valine (1.66 and 2.81) when compared to leucine (1.44 and 2.4) (Figure 1 and Figure 2) [29,30]. What is consistent throughout literature is the strength of the collective BCAAs for T2DM.

A consistent observation between valine, leucine and isoleucine is a deviation from the above-described trend in association in the intermediary temporal follow-up subgroup assessed (>6 to <12 years follow-up) (Figure 1, Figure 2 and Figure 3). Statistical significance was achieved in all assessed subgroups specific to each BCAA, potentially reflecting a non-linear relationship between T2DM incidence and temporality with respect to the mechanistic involvement of BCAAs. A temporally stratified assessment of BCAA plasma levels are seldom reported in either case–control or cohort studies, limiting comparisons with the existing literature [15,25]. Specifically, the majority of studies only report initial and final individual or total results, dependent on the duration of follow-up. As a result, a summative appraisal of the temporal trends observed in each of the BCAAs and their association with T2DM requires further investigation.

Recent studies have suggested that defects in the catabolic pathway of BCAAs may also be responsible for the accumulation of BCAAs in the plasma [50,51]. We therefore aimed to also appraise downstream BCAA metabolites and their association with T2DM over time. Unfortunately, insufficient data were available to statistically appraise BCAA metabolites, as four of the included studies did not report these, while the remaining studies all reported on different metabolites [29,31,41,45]. As evidenced by the robust and consistent association between BCAAs and T2DM, further research into determining the role of BCAA downstream metabolites in association with T2DM, and in conjunction with their upstream sources, is warranted.

### 4.2. Limitations

Despite the statistically significant and consistent results generated through the described approach, this study is not without limitations. Substantial inter-study heterogeneity was observed in the statistical analysis, which were likely due to demographic differences between the assessed populations. This likely also would explain the unusual trend observed in the 6–12-year sub-group. Due to a paucity in the available data, this current analysis was constrained by the inclusion of outcomes from different ethnic backgrounds, age distributions and follow-up durations. In addition, both targeted and untargeted metabolomic approaches were utilised by studies, which may also contribute to the heterogeneity, observed. Further, sources of heterogeneity were also observed per study with Palmer et al., 2015 yielded consistently for each individual BCAA analysis [29]. Equivocal result for analysis in the majority of temporal subgroups (Figure 1, Figure 2 and Figure 3) was observed. In addition, potential confounding factors were present in the included studies, despite attempt to control for BMI, diabetic status and subgroup per study design. Although, our sensitivity analysis confirmed the feasibility in including studies with a reduced independent variable adjustment count, the differences in covariate adjustment implemented by each of the incorporated studies may explain the substantial variation in confidence intervals observed across several included studies. This may be resolved in future studies through the obtaining of crude plasma BCAA levels for the assessed populations, with post hoc adjustment undertaken thereafter. Moreover, the extracted correlation data was insufficient for either qualitatively or quantitatively appraisal in relation to the presented findings (Table 4).

Through this systematic review and meta-analysis, we demonstrated consistent associations between each BCAA and the eventual development of T2DM irrespective of follow-up period duration. Due to the implemented study design and the absence of individual-level participant data, we were unable to explore any relationships between each of the BCAAs with disease severity, the trait-pharmacology interactions, or any clinically recognized subtypes, such as gestational or monogenic diabetes. Indeed, a recent metabolomic study (Del Coco et al., 2019) validated the existence of relationship heterogeneity between BCAAs and manifestations of diabetes mellitus beyond a recent-onset, metabolic syndrome implicated clinical presentation. [52] We therefore recommend further investigation into the specific associations between the recognized subtypes of diabetes mellitus, with the consideration of time as a separate variable, in future work.

## 5. Conclusions

In summation, the findings from this meta-analysis demonstrate that alterations in Plasma BCAA concentrations are associated with T2DM incidence, independent of the baseline plasma BCAA levels. We suggest the potential utility of BCAAs as biomarkers, which reflect early changes in T2DM. Detection of these changes years before the onset of physiological symptoms may be crucial for the early diagnosis and development of novel interventions.

## Figures and Tables

**Figure 1 nutrients-14-04411-f001:**
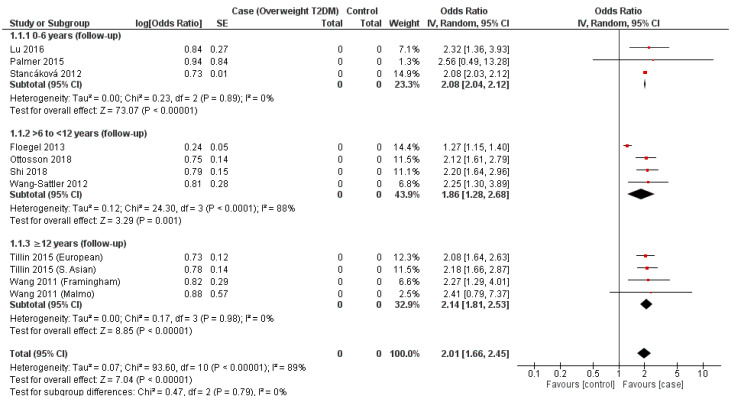
Forest plot depicting the effect sizes of the individual studies for valine in each of the designated temporal subgroups [29,30,31,41,42,43,44,45,46].

**Figure 2 nutrients-14-04411-f002:**
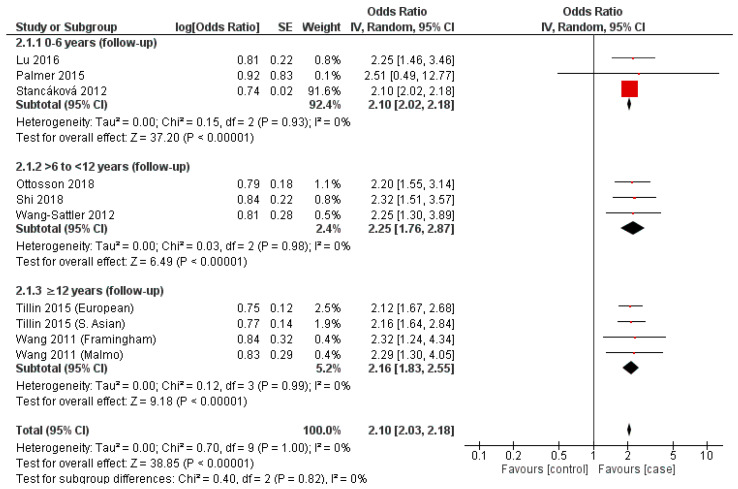
Forest plot depicting the effect sizes of the individual studies for leucine in each of the designated temporal subgroups [29,30,31,41,42,43,44,45,46].

**Figure 3 nutrients-14-04411-f003:**
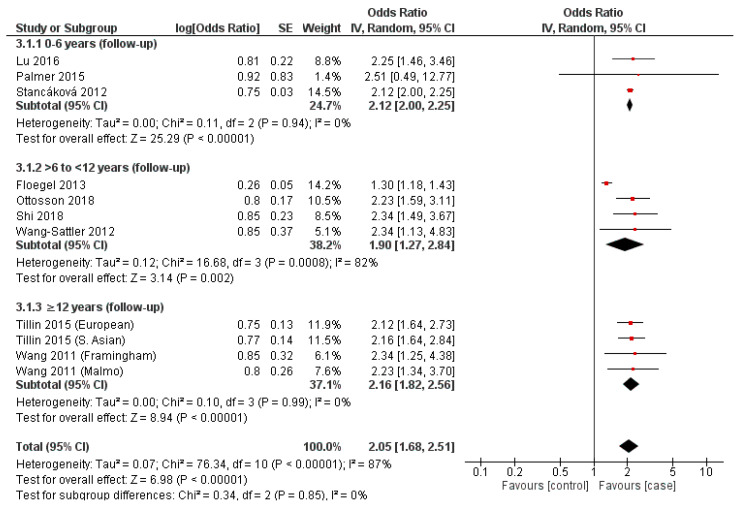
Forest plot depicting the effect sizes of the individual studies for isoleucine in each of the designated temporal subgroups [29,30,31,41,42,43,44,45,46].

**Table 1 nutrients-14-04411-t001:** Study characteristics of the manuscripts, which reached synthesis.

Lead Author	Publication Date	Study Design	Cases (n)	Control (n)	Patient Demographics (Intervention(s))	Follow-Up Period (Y)	Covariates	Cases Group Status	Control Group Status	Individual BCAAs	BCAA Metabolites
F Ottosson [31]	2018	Case-control	204	496	Mean of 69.5 years, predominantly male (69%), Swedish nationality	6.3	Age and Sex.	T2DM	Non-diabetic	Leucine, Isoleucine, Valine	C5, C4 and C3
Y Lu [30]	2016	Case-control	197	197	55.15 ± 2.8 years, predominantly female (59.4%), Chinese nationality	6	BMI, smoking, history of hypertension	T2DM	Non-diabetic,	Leucine, Isoleucine, Valine	C10
L Shi [41]	2018	Case-control	503	503	50.1 ± 8 years, predominantly female (55.5%), Swedish nationality	10	BMI, FBC, PA, education, smoking, consumption of alcohol, dietary fibre, red and processed meat, and coffee, plasma total cholesterol, triglycerols, and systolic and diastolic BP.	T2DM (no medication	Non-diabetic,	Leucine, Isoleucine, Valine	C3 and KMV
T Wang (Framingham) [42]	2011	Case-control	189	189	56.5 ± 8.5 years, predominantly male (58%), USA nationality	12	Age, sex, BMI, fasting glucose, and parental history.	T2DM	Non-diabetic	Leucine, Isoleucine, Valine	-
T Wang (Malmo) [42]	2011	Case-control	163	163	Mean of 58 years, predominantly female (55%), Swedish nationality	12.6	Age, sex, BMI, and fasting glucose.	T2DM	Non-diabetic	Leucine, Isoleucine, Valine	-
A Floegel [43]	2013	Case-control	800	2282	52.1 ± 8.1 years, predominantly women (52.1%), German nationality	7	age, sex, alcohol intake, smoking, physical activity, education, coffee intake, red meat intake, prevalent hypertension, BMI, and waist circumference (cm)	T2DM	Non-diabetic	Isoleucine, Valine	-
R Wang-Sattler [44]	2012	Case-control	91	866	64.7 ± 5.45 years, predominantly male (53%), German nationality	7	Age, sex, BMI, physical activity, alcohol intake, smoking, systolic BP, HDL cholesterol, HbA1c, fasting glucose and fasting insulin	T2DM(no medication)	Non-diabetic	Leucine, Isoleucine, Valine	C2
A Stancáková [45]	2012	Case-control	646	3026	57 ± 7 years, all male, Finnish nationality	4.7	Age and BMI	T2DM(no medication)	Non-diabetic	Leucine, Isoleucine, Valine	-
T Tillin (European) [46]	2015	Case-control	643	1007	50.6 ± 7.0 years, all male, South Asian origin	19	Age, WHR, truncal skinfold thickness, Matsuda-IR, HDL cholesterol level, current smoking, and alcohol consumption.	T2DM(no medication)	Non-diabetic	Leucine, Isoleucine, Valine	-
T Tillin (South Asian) [46]	2015	Case-control	801	1279	52.75 ± 7.25 years, all male, European origin	19	Age, WHR, truncal skinfold thickness, Matsuda-IR, HDL cholesterol level, current smoking, and alcohol consumption.	T2DM(no medication)	Non-diabetic	Leucine, Isoleucine, Valine	-
ND Palmer [29]	2015	Case-control	76	70	56 ± 8 years, predominantly female (63%), European-American, Hispanic, and African American ethnicity	5	Age, sex, and BMI	T2DM(no medication)	Non-diabetic	Leucine or isoleucine, valine	C2, C5 and C10

T2DM = type 2 diabetes mellitus, BCAAs = branched chain amino acids, C2 = Acetylcarnitine, C3 = Propionylcarnitine, C4 = Butyrylcarnitine, C5 = Isovalerylcarnitine, C10 = Decenoylcarnitine, KMV = 3-methyl-2-oxovaleric acid, - no results, BMI = body mass index, FBC = fasting blood glucose, PA = physical acitivity, WHR = waist hip ratio, IR = insulin resistance, BP = blood pressure, HDL = high-density lipid.

**Table 2 nutrients-14-04411-t002:** Calculated log(OR) for individual serum BCAA levels per each data-point incorporated into the meta assessment.

Lead Author	Publication Date	Valine Outcome Measure			Leucine Outcome Measure			Isoleucine Outcome Measure		
		Log(OR)	SE	*p*-Value	Log(OR)	SE	*p*-Value	Log(OR)	SE	*p*-Value
F Ottosson [31]	2018	0.73	0.01	0.94	0.74	0.02	0.197	0.75	0.03	0.064
Y Lu [30]	2016	0.94	0.84	0.0003 *	0.92	0.83	0.0023	0.92	0.83	0.4 *
L Shi [41]	2018	0.84	0.27	<0.001 *	0.81	0.22	0.002	0.81	0.22	0.002 *
T Wang [42]	2011	0.75	0.14	0.34	0.79	0.18	0.034	0.80	0.17	0.01 *
T Wang [42]	2011	0.24	0.05	5.89 × 10^−5^ *	-	-	-	0.26	0.05	3.04 × 10^−5^ *
A Floegel [43]	2013	0.81	0.28	0.03 *	0.81	0.28	0.06	0.85	0.37	0.008 *
R Wang-Sattler [44]	2012	0.79	0.15	0.016 *	0.84	0.22	0.006	0.85	0.23	0.001 *
A Stancáková [45]	2012	0.82	0.29	0.02 *	0.84	0.32	0.006	0.85	0.32	0.004 *
T Tillin [46]	2015	0.88	0.57	0.01 *	0.83	0.29	0.009	0.80	0.26	0.09
T Tillin [46]	2015	0.78	0.14	0.044 *	0.77	0.14	0.074	0.77	0.14	0.13
ND Palmer [29]	2015	0.73	0.12	0.9	0.75	0.12	0.4	0.75	0.13	0.4

OR = Odds Ratio, SE = Standard Error, * = Statisticall significant *p* < 0.05, - = No data available.

**Table 3 nutrients-14-04411-t003:** Sensitivity analysis of all available datapoints, per temporal subgroup, where model adjustment with less than five covariates was undertaken by the publishing team.

Temporal Subgroup (Years Follow-Up)	Data Sources (Lead Author; Year of Publication; BCAAs)	Covariates (Lead Author; Variables)	OR	95% CI (Lower–Upper)	*p*-Value
0 to 6	Palmer, 2015 [29] (iso, leu, val) Lu, 2016 [30] (iso, leu, val)	Palmer; Age, sex, BMI, and AIR Lu; BMI, smoking, history of hypertension	2.28	1.77–2.94	*p* < 0.00001
>6 to <12	Ottosson, 2018 [31] (iso, leu, val)	Ottosson; age and sex	2.17	1.81–2.60	*p* < 0.00001
≥12	Wang (Malmo dataset), 2011 [42] (iso, leu, val)	Wang [Malmo dataset]; age, sex, BMI, and fasting glucose	2.27	1.59–3.25	*p* < 0.00001

Legend Iso = Isoleucine, leu = leucine, val = valine, BMI = body mass index, AIR = acute insulin response.

**Table 4 nutrients-14-04411-t004:** Presented correlation statistics between serum BCAA levels with either of body mass index, glycated haemoglobin, insulin resistance and/or fasting glucose in the assessed studies at baseline.

Lead Author	Valine						Leucine						Isoleucine					
	BMI		HOMA-IR		HbA1c		BMI		HOMA-IR		HbA1c		BMI		HOMA-IR		HbA1c	
	Outcome	*p*-Value	Outcome	*p*-Value	Outcome	*p*-Value	Outcome	*p*-Value	Outcome	*p*-Value	Outcome	*p*-Value	Outcome	*p*-Value	Outcome	*p*-Value	Outcome	*p*-Value
F Ottosson [31]	-	-	-	-	-	-	-	-	-	-	-	-	-	-	-	-	-	-
Y Lu [30]	-	-	-	-	-	-	-	-	-	-	-	-	-	-	-	-	-	-
L Shi (S) [41]	0.24	-	0.2	0.01	-	-	0.26	-	0.25	<0.001	-	-	0.26	-	0.29	<0.001	-	-
T Wang (P) (Framingham) [42]	-	-	0.24	0.0008	-	-	-	-	0.24	0.0009	-	-	-	-	0.24	0.0007	-	-
T Wang (Malmo) [42]	-	-	-	-	-	-	-	-	-	-	-	-	-	-	-	-	-	-
A Floegel [43]	-	-	-	-	-	-	-	-	-	-	-	-	-	-	-	-	-	-
R Wang-Sattler (P) [44]	0.27	-	-	-	0.08	-	0.19	-	-	-	0.09	-	0.19	-	-	-	0.09	-
A Stancáková [45]	-	-	-	-	-	-	-	-	-	-	-	-	-	-	-	-	-	-
T Tillin (S) (European) [46]	0.22	<0.05	0.27	<0.05	-	-	0.18	<0.05	0.26	<0.05	-	-	0.23	<0.05	0.34	<0.05	-	-
T Tillin (S) (South.Asian) [46]	0.29	<0.05	0.33	<0.05	-	-	0.27	<0.05	0.31	<0.05	-	-	0.31	<0.05	0.35	<0.05	-	-
ND Palmer [29]	-	-	-	-	-	-	-	-	-	-	-	-	-	-	-	-	-	-

S = Spearmans correlation test, P = Pearsons correlation test, BMI = body mass index.

## Data Availability

The data presented in this study are available on request from the corresponding author.

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
