# Peer review of "The Association between Circulating Branched Chain Amino Acids and the Temporal Risk of Developing Type 2 Diabetes Mellitus: A Systematic Review & Meta-Analysis"

_nutrients, 2022, doi:10.3390/nu14204411_

Round 1
Reviewer 1 Report
the authors present the results of a met-analysis aimed at exploring whether BCAA levels are prospectively associated with the development of T2DM: Overall, the structure of the systematic review is appropriate and the results are clear also to the non-expert reader. I have some minor comments to improve the clarity and appropriateness of the manuscript:
1- The protocol has been deposited in Prospero but its review is still pending. Looking at the dates, I noted that the search was terminated (November) before loading the protocol on Prospero (January). This is OK but I would state this aspect in the methods, since the initial question should be not modified by the results of the search (e.g. the time-dependent analysis must have been pre-specified and not an idea derived from the search).
2- Much of the methods are presented in supplementary files. While this might have been prompt by space limits, some passages should be necessarily repeated in the main text. Otherwise the reader would be confused about the work performed by the authors. For instance, inclusion/exclusion criteria for manuscript inclusion are not reported. Where only prospective studies considered? Cross-sectional studies were excluded? How such temporal windows for categorization were chosen?There was a minimum cut-off length? Why in the title is reported obesity-induced diabetes? Do all the patients enrolled were obese? What about diabetes without obesity (which involve a consistent number of subjects)? Please report at least some of these details in the methods and try to be as specific as possible.
2- Summary measures are shown in tables but the forest plot used to generate them is not displayed. In this way, the relative weight of each paper can not be estimated. For instance, if one paper covers the 90% of the subjects enrolled this is not observable in this manner. Please show (in the main manuscript or in the supplementary) the relative forest plots.
3- No reasons for exclusion of the non-collected manuscript is mentioned in the text. Albeit this is visible on the supplementary, at least the manuscripts reaching the final stage should be discussed (i.e. the reasons leading to their exclusion should be commented in this section of the results). Similarly, the characteristics of the patients included in the final analysis are not described. Which was the percentage of male sex in the overall population? The prevalence of obesity? The mean age? Please use all the info collected to present a weighted mean of the characteristics of the population resulting from the 9 papers.
4- The categorization for "temporal windows" (i.e. follow-up duration) is interesting but reduces the number of papers included in each analysis, limiting the resulting power. Thus, I would perform and show also at least one analysis pooling all the results together to THEN show temporally stratified results.
5- (minor). Albeit not included in the main analysis due to lack of fitting with inclusion criteria, much more papers have tried to explore a relationship of BCAA with T2DM. For example, here (DOI: 10.3390/jcm8050720) the authors obtained unexpected results showing that patients with advanced diabetes have lower (and not higher) levels of BCAA compared to both healthy subjects and patients with less severe diabetes. This implies that the stage of the disease can affect the trend of the observed difference. Other papers also found an inverse (rather than positive) association. Please comment in the discussion section.
Reviewer 2 Report
This is a paper reporting results of a systematic review and meta-analysis examining the relationship between BCAAs and incident T2D. The topic is of great interest as the access to metabolomic techniques will most probably increase in the near future and BCAAs are good candidates for risk biomarkers for T2D.
There are few clarification sought to be needed to better understand the results. In Material and Methods section, the Eligibility criteria, although described in Supplementary materials, should include the main information regarding the type of studies included. It is not clear which were the time points when BCAAs were measured throughout the studies. The authors state that "Plasma BCAA concentrations are associated with T2DM
incidence, independent of the baseline plasma BCAA levels " but the Results section is it unclear in this respect.
The authors also state that "the risk of developing T2DM with differentials in the respective follow-up times of 0-6 years, 6-12 years and ≥12 years follow-up for valine (OR=2.08, 1.86 & 2.14), leucine (OR=2.10, 2.25 & 2.16) and isoleucine (OR=2.12, 1.90 & 2.16) is demonstrated" but this is not very clear from the Results section. Was statistical significance tested to compare ORs in the three categories of follow-up periods?
Figure 1 is missing from the submitted documents and this makes the understanding of the results more difficult.
